

# Drought-induced reduction in methane fluxes and its hydrothermal sensitivity in alpine peatland

Haidong Wu[1,2,3,*], Liang Yan[1,2,3,*], Yong Li[1,2,3], Kerou Zhang[1,2,3], Yanbin Hao[4], Jinzhi Wang[1,2,3], Xiaodong Zhang[1,2,3], Zhongqing Yan[1,2,3], Yuan Zhang[4] and Xiaoming Kang[1,2,3]

[1] Institute of Wetland Research, Chinese Academy of Forestry, Beijing, China
[2] Beijing Key Laboratory of Wetland Services and Restoration, Beijing, China
[3] Sichuan Zoige Wetland Ecosystem Research Station, Tibetan Autonomous Prefecture of Aba, China
[4] University of Chinese Academy of Science, Beijing, China
[*] These authors contributed equally to this work.

Corresponding author
Xiaoming Kang, xmkang@ucas.ac.cn

## ABSTRACT

Accurate estimation of $CH_4$ fluxes in alpine peatland of the Qinghai-Tibetan Plateau under extreme drought is vital for understanding the global carbon cycle and predicting future climate change. However, studies on the impacts of extreme drought on peatland $CH_4$ fluxes are limited. To study the effects of extreme drought on $CH_4$ fluxes of the Zoige alpine peatland ecosystem, the $CH_4$ fluxes during both extreme drought treatment (D) and control treatment (CK) were monitored using a static enclosed chamber in a control platform of extreme drought. The results showed that extreme drought significantly decreased $CH_4$ fluxes in the Zoige alpine peatland by 31.54% ($P < 0.05$). Extreme drought significantly reduced the soil water content (SWC) ($P < 0.05$), but had no significant effect on soil temperature (Ts). Under extreme drought and control treatments, there was a significant negative correlation between $CH_4$ fluxes and environmental factors (Ts and SWC), except Ts, at a depth of 5cm ($P < 0.05$). Extreme drought reduced the correlation between $CH_4$ fluxes and environmental factors and significantly weakened the sensitivity of $CH_4$ fluxes to SWC ($P < 0.01$). Moreover, it was found that the correlation between subsoil (20 cm) environmental factors and $CH_4$ fluxes was higher than with the topsoil (5, 10 cm) environmental factors under the control and extreme drought treatments. These results provide a better understanding of the extreme drought effects on $CH_4$ fluxes of alpine peatland, and their hydrothermal impact factors, which provides a reliable reference for peatland protection and management.

## INTRODUCTION

In recent years, due to the aggravation caused by human activities, the global atmospheric and water cycle pattern has been significantly changed, resulting in an increasing frequency and intensity of global extreme climate events (*IPCC, 2013*; *Kreyling et al., 2008*; *Kang et al., 2018*; *Thakur et al., 2017*). Recent studies have indicated that the occurrence of extreme

drought events can significantly change the water and heat conditions of the ecosystem, affecting the physiological state of plants and activities of soil microbes, triggering changes in the soil structure and function, and breaking the original carbon balance of the ecosystem, which in turn can aggravate the intensity and frequency of extreme drought events on a global scale (*Reichstein et al., 2013*; *Bloor & Bardgett, 2012*; *Beierkuhnlein et al., 2011*; *Hao et al., 2011*). However, research on extreme drought is still concentrated in arid and semi-arid grasslands at present, and research on peatland is relatively rare (*Bloor & Bardgett, 2012*; *Beierkuhnlein et al., 2011*; *Yin et al., 2013*). As an important global carbon pool, peatlands are carbon-rich ecosystems that cover just 3% of the Earth's land surface, but they store one-third of the soil carbon (*Dargie et al., 2017*; *Page, Rieley & Banks, 2011*). As such, peatlands play an important role in the global carbon cycle and mitigation of climate change (*Wu, 2012*).

The alpine peatland ecosystem, on account of its special altitude, presentsa higher sensitivity to climate change (*Mclaughlin & Webster, 2014*). Additionally, with low temperatures and anoxia all year round, peatlands have sequestered large amounts of carbon in the soil (*Van Bellen, Garneau & Booth, 2011*; *Bunbury, Finkelstein & Bollmann, 2012*). However, when disturbed by external conditions, the source and sink of $CH_4$ in the alpine peatland ecosystem can be significantly altered (*Webster et al., 2013*). As one of the main greenhouse gases, the warming potential of $CH_4$ is 23 times than that of $CO_2$, and changes in the $CH_4$ content in the atmosphere can have a significant impact on the trend and intensity of global climate change (*Reichstein et al., 2013*; *Soren, Sejian & Malik, 2015*). However, the dynamics of $CH_4$ in alpine peatland ecosystems and its response to extreme drought are poorly understood and lack quantified analyses. Therefore, accurate quantification of alpine peatland $CH_4$ fluxes under extreme drought conditions at various spatial and temporal scales is crucial and necessary for fully understanding the climate change process.

The Zoige plateau, located in the northeast of the Qinghai-Tibet plateau, is the region with the highest organic carbon reserves in China and one of the largest plateau peatlands in the world, thus playing an important role in the global carbon cycle (*Wang et al., 2012*). As such, this region could potentially have a significant impact on regional climate change (*Kang et al., 2014*). However, due to the warming and drying trends that have occurred over the past 30 years, the surface water level of the Zoige peatland has decreased substantially, which directly alters the pattern of $CH_4$ fluxes in this area (*Rydin & Jeglum, 2006*; *Yang et al., 2014*; *Gorham, 1991*; *Chen et al., 2013*). Moreover, changes in precipitation and atmospheric temperatures, as well as the effects of decreased water levels, serve to increase the level of uncertainty regarding the magnitude of the $CH_4$ fluxes occurring in many ecosystems (*Chen et al., 2013*; *Blankinship et al., 2010*). Therefore, to improve our understanding of the $CH_4$ dynamics occurring in the Zoige alpine peatland, the effects of temperature and precipitation variability under extreme drought conditions should be studied simultaneously.

In recent years, researchers have found that $CH_4$ uptake is strongly controlled by soil moisture, as soil temperature only has a minor influence on $CH_4$ fluxes measured at the Tasmania Ecological Research site (*Fest et al., 2017*). Daily observations of $CH_4$ fluxes in

nine different types of swamps in northern Finland have shown that the average $CH_4$ emission is significantly correlated with the average groundwater level (*Huttunen et al., 2003*). Related research has also indicated that the yield of $CH_4$ is lower under drought conditions in peatlands (*Freeman et al., 2002*). Moreover, frequent extreme drought events in recent years have been increasing, and these events have clearly had a profound impact on $CH_4$ fluxes in the Zoige peatland (*Chen et al., 2009*; *Wang, Ding & Wang, 2003*). However, data regarding the changes in $CH_4$ fluxes in the Zoige peatland under extreme drought are limited.

Therefore, the accurate estimation of $CH_4$ fluxes and the factors impacting their dynamics will help quantify the interactions and feedback occurring between extreme drought events and the alpine peatland ecosystem. In this study, we observed the $CH_4$ fluxes and environmental factors at the Zoige peatland in a controlled experiment of extreme drought with the hope of estimating the drought effects on $CH_4$ fluxes, and we identified the environmental variables affecting these fluxes under continuous drought stress. The results provide an important scientific basis to accurately evaluate the contribution of alpine peatland $CH_4$ towards global climate change and will also help support peatland conservation.

## MATERIALS & METHODS

### Site description

The experiment was conducted in Zoige county in the eastern Tibetan Plateau (33.79°N, 102.95°E) at an altitude of 3,430 m (Fig. 1A). The mean annual temperature is 1.1 °C, and mean annual precipitation is 648.5 mm, with 80% falling during the growing season from June to September. The mean monthly temperature ranges from 1 °C (January) to 11 °C (July). The experiment was established in a frigid temperate zone steppe dominated by herbaceous marshes and composed mainly of *Carex meyeriana*, *Carex muliensis*, and *Kobresia tibetica*. The main soil type was marshy peat, with the soil pH is between 6.8–7.2 in localized areas (*Zhou et al., 2015*). The depth of peat in the vertical profile of this site is in general 1.2 m. Field experiments were approved by the Institute of Wetland Research.

### Experiment design and data collection

Based on the local rainfall data for the past 50 years, we defined daily rainfall ≥ three mm as ecologically effective precipitation (*Hao et al., 2012*). During the flourishing period of the growth season, we selected 32 days as the duration of non-ecologically effective precipitation (drought days) and simulated extreme drought over this period of plant growth (*Wang et al., 2007*). The area of the plot was 20 m × 20 m, and extreme drought treatment (D) and control treatments (CK) were set up, independently, with each treatment consisting of three (2 m × 2 m) repetition plots (Fig. 1B). We buried iron sheets in the soil about 1 m deep around each treatment to prevent the lateral flow of soil water. A stainless-steel base (50 cm × 50 cm × 20 cm) was placed at the sampling point and inserted into the ground at a depth of 10 cm. Before each measurement, we filled the groove of stainless steel with water to ensure the airtightness of the measurement (Fig. 1C). For the extreme drought treatment, we used a magnesium-aluminum alloy shelter (length × width

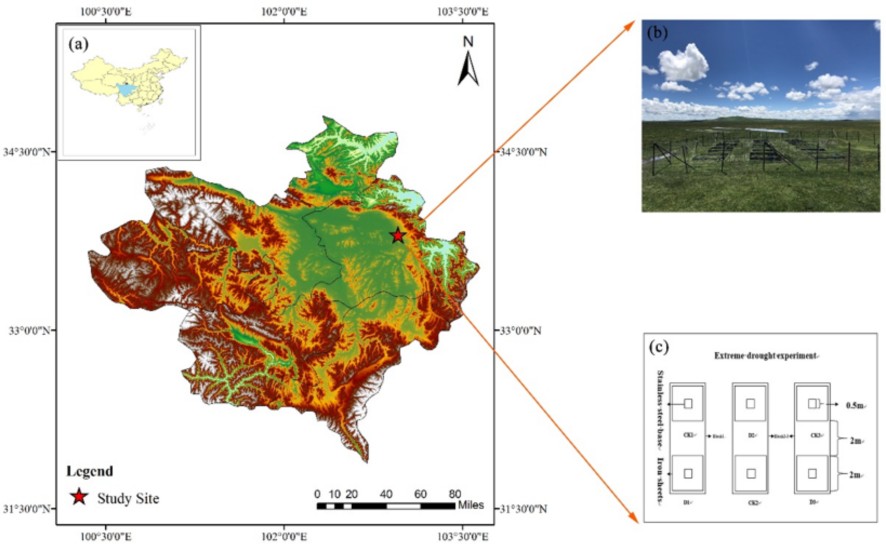

**Figure 1** (A) Zoige peatland in the eastern part of the Tibetan Plateau with the location of the study site, Sichuan province; (B) the picture of experiment site; (C) the zoning schematic map of experiment plot.

× height; 2.5 m × 2.5 m × 1.8 m) to simulate drought, and the light transmittance of the shelter was more than 90%. The gas in the controlled plot was monitored under natural conditions.

A fast greenhouse gas analyzer (DLT-100, Los Gatos Research, USA) was used to monitor $CH_4$ fluxes, at a data acquisition frequency of 1 Hz. A TZS-5X thermometer was used to monitor the air temperature (Ta) and soil temperature (Ts), and a TDR 300 was used to measure the SWC. A box (50 cm × 50 cm) was connected with the fast greenhouse gas analyzer. There were two small holes two cm in diameter at the top of the box, which were closed with rubber plugs. There was a small hole in each rubber plug for the insertion of two gas conduits (intake pipe and outlet pipe) with a length of 20 m and an inner diameter about four mm. The box was connected to an intake pipe and an outlet pipe with a length of about 20 m. To ensure the gas in the box could be quickly mixed and evenly distributed, two small fans (10 cm in diameter) were set at the top of the box. Each sampling point was measured in a sealed transparent box or dark box for 2 min, and the measured data from the dark box were used to ascertain the $CH_4$ fluxes. The drought treatment started on July 15, 2017, and end on August 16, 2017. The measurements were taken at three periods of one day (first: 9:00–10:00, second: 12:00–13:00, third: 14:00–15:00). For the measurement of aboveground biomass, 50 cm × 50 cm quadrats were randomly chosen in each experimental plot, and all plants within the quadrats were cut to ground level. After the dust was removed, the plant material was oven dried to constant weight at 70 °C. Belowground biomass was collected by digging soil pits at the same locations where the aboveground biomass had been removed at the sampling depths of 0–20 cm and 20–40 cm. Soils containing root biomass were placed in 40-mesh nylon bags and taken back to the

laboratory, where the roots were carefully washed and then oven dried to a constant weight at 70 °C. A soil drill was used to sample the soil via multi-point sampling and mixing. The soil organic matter (SOC) was determined using a potassium dichromate volumetric method (*Wang et al., 2007*), total carbon (TC) was determined by the elemental analyzer (*Sokolova & Vorozhtsov, 2014*), and total nitrogen (TN) was determined via the Kjeldahl method (*Nozawa et al., 2005*).

## Data analysis

The formula used for calculating the greenhouse gas fluxes (*Wickland et al., 2001*) was:

$$F_c = \frac{\partial C'}{\partial t} \times \frac{M}{V_0} \times \frac{P}{P_0} \times \frac{T_0}{T_0 + t} \times \frac{H}{100} \times 3600 \tag{1}$$

where $F_c$ is the gas fluxes (mg C/ (m2 h)); M is the molar mass of gas (g/mol); $V_0$ is the standard molar volume of gas (22.4 L/mol); $P/P_0$ is the measurement of pressure to standard air pressure; $T_0$ is the absolute temperature (273.15 °C); t is the average value of the measured temperature in the box (°C); and H is the static height (cm). Importantly, the measured data were analyzed by linear regression to calculate the linear slope of the gas concentration relative to the time of observation.

Repeated-measure ANOVA with Duncan's multiple-range tests were performed to examine the main and interaction effects of date, treatment and block on the differences in $CH_4$ fluxes and environmental factors in 2017 (SPSS, Chicago, IL, USA). A one-way ANOVA analysis was performed to examine the properties (above and below ground biomass, TC, TN, SOC) at different depths in 2017 (SPSS, Chicago, IL, USA). To further evaluate the relationship of $CH_4$ fluxes and environmental factors, a correlation matrix analysis between $CH_4$ fluxes and environmental factors was conducted (Origin 2017, USA). The slopes of those linear relationships were analyzed and compared by SMA (Standardized Major Axis) regression analysis, using the SMATR (Standardized Major Axis Tests and Routines) package (*Warton et al., 2006*). R v3.5.1 with the corrplot package was used for the correlation analysis (*Svetnik et al., 2004*).

## RESULTS

### Climate during the experiment period

During the experiment period (32 d), 14 precipitation events occurred in Zoige, with 6 days including ecologically effective precipitation events (≥3 mm). The daily precipitation ranged from 0.1 mm to 20.6 mm (Fig. 2) and the average precipitation was 1.9 mm. The total precipitation was 58.9 mm in the control treatment and 0 mm in the extreme drought treatment during the experimental period. The precipitation mainly occurred in early August, and a transient rainfall occurred at the end of the treatment period. The highest and lowest daily temperatures were 15.3 °C and 8.4 °C, respectively, and the average temperature was 12.9 °C during the treatment period.

### Effects of extreme drought on $CH_4$ fluxes

From the end of June to the middle of July, there was a transition period between a weak $CH_4$ sink and a weak $CH_4$ source of the Zoige peatland (Fig. 3A). The emission of $CH_4$

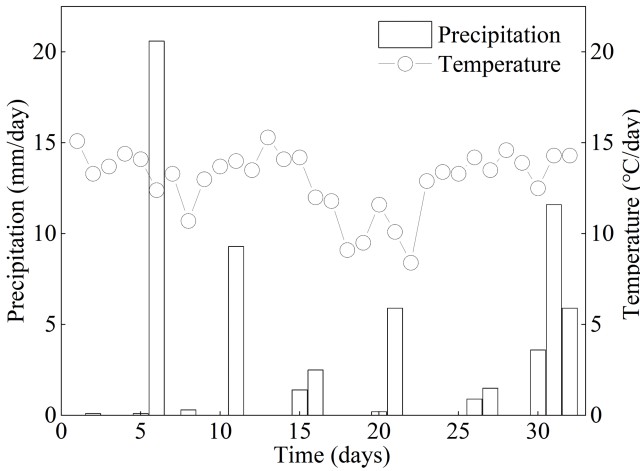

**Figure 2  Daily average precipitation and temperature of Zoige peatland during the experimental period in 2017.** Point-line chart and histogram indicate temperature and precipitation, respectively.

from the Zoige peatland reached a maximum around August 16. During the pre-drought period, the ecosystem functioned as a $CH_4$ sink, and during the extreme drought and post-drought periods, the ecosystem functioned as a net $CH_4$ source (Fig. 3B). Compared to the control treatment, extreme drought significantly decreased the $CH_4$ fluxes of the Zoige peatland ecosystem by 31.54% ($P < 0.05$, Fig. 3B, Table 1) in the drought period, and there was no significant change in the pre and post-drought periods of the experiment under the extreme drought and control treatment ($P > 0.05$). Additionally, the difference of $CH_4$ fluxes between the control and drought reached the highest value at the peak of plant growth (Fig. 3C). The extreme drought significantly decreased SWC at depths of 5, 10, and 20 cm ($P < 0.05$), but there was no significant influence of the extreme drought on Ts at depths of 5, 10, or 20 cm ($P > 0.05$, Table 1).

## Effects of extreme drought on plant biomass and soil physicochemical properties

The extreme drought treatment significantly decreased the aboveground biomass of the Zoige alpine peatland ecosystem by 42.75% ($P < 0.05$, Fig. 4A). The extreme drought treatment significantly decreased the belowground biomass by 59.73% and 59.65% at a depth of 0–10 cm and 10–20 cm, respectively ($P < 0.05$, Fig. 4B). Under both treatments, the root mass of the subsoil (10–20 cm) was higher than that of the topsoil (0–10 cm) (Fig. 4B). Subsoil (20 cm) SWC was higher than that of the topsoil (5, 10 cm) (Fig. 4C). Significant differences in TC and TN between the two treatments were observed at a depth of 10–20 cm ($P < 0.05$, Figs. 4D–4E), but there was no significant difference in TC or TN between the extreme drought treatment and control treatment at depths of 0–10 cm ($P > 0.05$, Figs. 4D–4E). There was also no significant difference in SOC at depths of 0–10 and 10–20 cm ($P > 0.05$, Fig. 4F). The organic matter (TC, TN, and SOC) of the subsoil was lower than that of the top soil (Figs. 4D–4F).

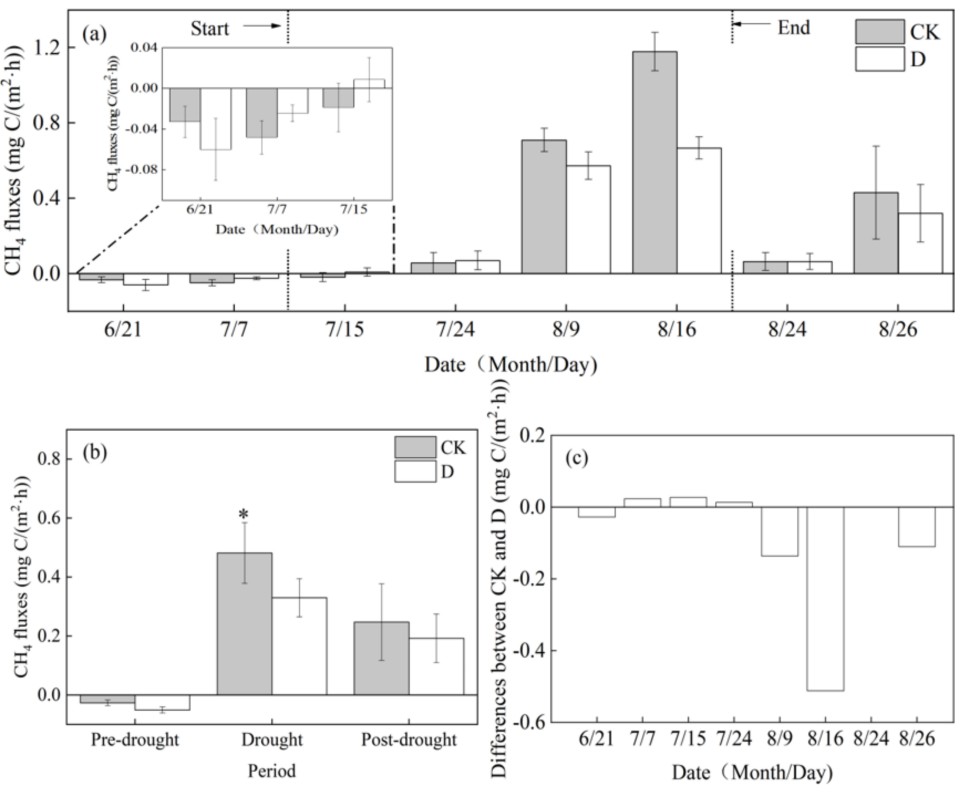

**Figure 3** (A) Effects of extreme drought on $CH_4$ fluxes in 2017; (B) the total mean value at different periods; (C) the difference value between the extreme drought and control treatments. Bars show ± SE ($n = 3$). The arrows indicate the dates of the experiment. *: statistically significant at $P < 0.05$. CK, control; D, extreme drought.

**Table 1** Results ($P$ value) of effects of CH4 fluxes, Ts5, Ts10, Ts20, SWC5, SWC10 and SWC20 on block, date, drought, date*drought and date*block in 2017. Ts 5/10/20, soil temperature at depth of 5, 10 and 20 cm; SWC 5/10/20, soil water content at depth of 5, 10 and 20 cm.

|  | CH₄ fluxes | Ts5 | Ts10 | Ts20 | SWC5 | SWC10 | SWC20 |
|---|---|---|---|---|---|---|---|
| Block | 0.679 | 0.960 | 0.999 | 0.900 | 0.072 | 0.066 | 0.034 |
| Date | <0.001 | <0.001 | <0.001 | <0.001 | <0.001 | <0.001 | <0.001 |
| Drought | 0.015 | 0.624 | 0.617 | 0.499 | 0.023 | 0.034 | 0.033 |
| Date*Drought | <0.001 | 0.775 | 0.960 | 0.937 | 0.354 | 0.883 | 0.499 |
| Date*Block | 0.006 | 0.623 | 0.994 | 0.947 | 0.100 | 0.790 | 0.793 |

## Relationship between $CH_4$ fluxes and environmental factors

The regression analysis showed that the Ts at the depth of 10 and 20 cm had a significantly negatively relationship with $CH_4$ fluxes between the two treatments ($P < 0.05$, Figs. 5B–5C), as the $CH_4$ fluxes gradually decreased as the Ts increased. The correlation between the subsoil (20 cm) temperature and $CH_4$ fluxes was higher than it was with the topsoil (5, 10 cm) temperature between the two treatments (Figs. 5A–5C). The dynamics of the $CH_4$ fluxes correlated well with that of the SWC, both in the extreme drought and control

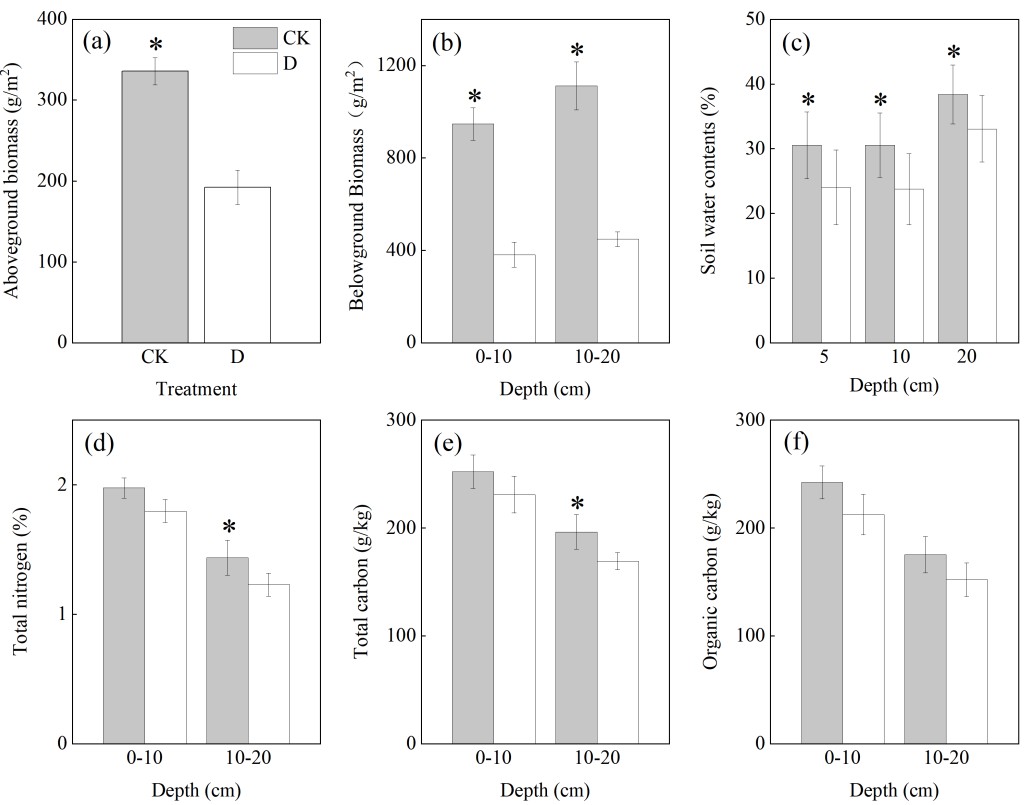

**Figure 4** (A) The impacts of extreme drought on aboveground biomass; (B) the impacts of extreme drought on belowground biomass; (C) the impacts of extreme drought on SWC at depths of 5, 10 and 20 cm; (D) the effects of extreme drought on total nitrogen in the different soil layers; (E) the effects of extreme drought on total carbon in the different soil layers; (F) the effects of extreme drought on soil organic carbon in the different soil layers.

treatments (Figs. 5D–5F). The SWC at depths of 5 ($P < 0.01$), 10 ($P < 0.01$), and 20 ($P < 0.01$) cm was negatively correlated with the $CH_4$ fluxes under the extreme drought and control treatments (Figs. 5D–5F). The correlation between the subsoil (20 cm) water content and $CH_4$ fluxes was higher than it was with the topsoil (5, 10 cm) water content between the two treatments. Moreover, there was a significant difference in the slopes of the SWC at depths of 5, 10, and 20 cm between the control and drought treatments ($P_{slope} < 0.01$, Figs. 5D–5F). The slope of the $CH_4$ fluxes under the extreme drought treatment was lower than that under the control treatment relative to the SWC. The correlation of $CH_4$ fluxes to SWC was higher than it was relative to Ts (Figs. 5A–5F).

The correlation matrix analysis between $CH_4$ fluxes and the different environmental factors at depths of 5, 10, and 20 cm were negative under the two treatments. The correlation between $CH_4$ fluxes and subsoil (20 cm) environmental factors (SWC and Ts) was higher than that with the topsoil (5, 10 cm) environmental factors (Figs. 6A–6D). The extreme drought decreased the correlation between the Ts and $CH_4$ fluxes (Figs. 6A–6B), and the extreme drought decreased the correlation between the SWC and $CH_4$ fluxes (Figs.

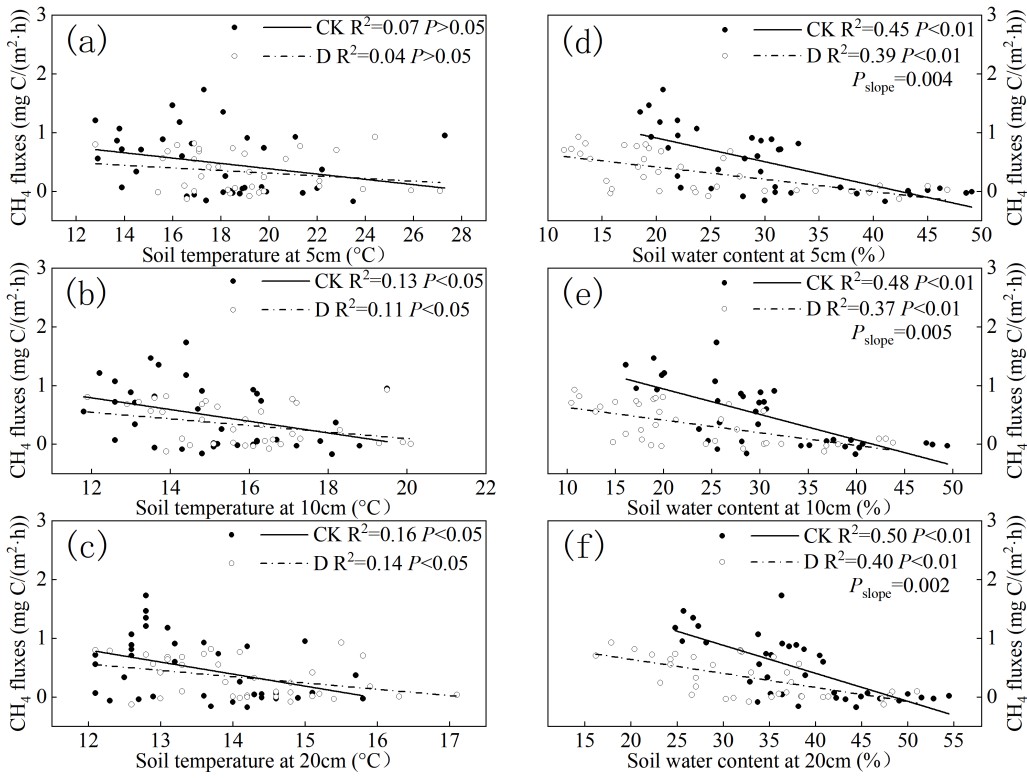

**Figure 5** **Relationships between CH$_4$ fluxes and (A) 5 cm, (B) 10 cm, and (C) 20 cm soil temperature, and the relationships between CH$_4$ fluxes and (D) 5 cm, (E) 10 cm, and (F) 20 cm SWC in the different treatments.** CK, control; D, extreme drought. $P < 0.05$ indicates a significant difference between CH$_4$ fluxes and environment factors (Ts, SWC). $P_{slope} < 0.05$ indicates a significant difference in the slopes between control and drought treatment.

6C–6D). There was a stronger relationship between the SWC and CH$_4$ fluxes than between the Ts and CH$_4$ fluxes (Figs. 6A–6D).

## DISCUSSION

The influence of extreme drought in relation to the variation of CH$_4$ fluxes has been recognized in earlier studies (*Wang, Ding & Wang, 2003*; *Harriss, Sebacher & Day, 1982*; *Borken et al., 2006*; *Stiehl-Braun et al., 2011*; *Hartmann, Buchmann & Niklaus, 2011*; *Goodrich et al., 2013*). For instance, CH$_4$ fluxes measured by the eddy covariance method at Mer Bleue bog in Canada suggested that the total CH$_4$ emitted during the growing season with extreme drought was less than that during the previous wetter year (*Brown et al., 2013*). Meanwhile, three drought scenarios (gradual, intermediate, and rapid transition into drought) at 18 freshwater wetlands investigated in Everglades National Park, USA revealed that more CH$_4$ was emitted than net carbon uptake could offset as the relative humidity increased (*Malone et al., 2013*). Our study used a control experiment to simulate an extreme drought event for the reason that the controlled experiment had better consistency in soil and vegetation conditions. We analyzed the effects of extreme drought on CH$_4$ fluxes

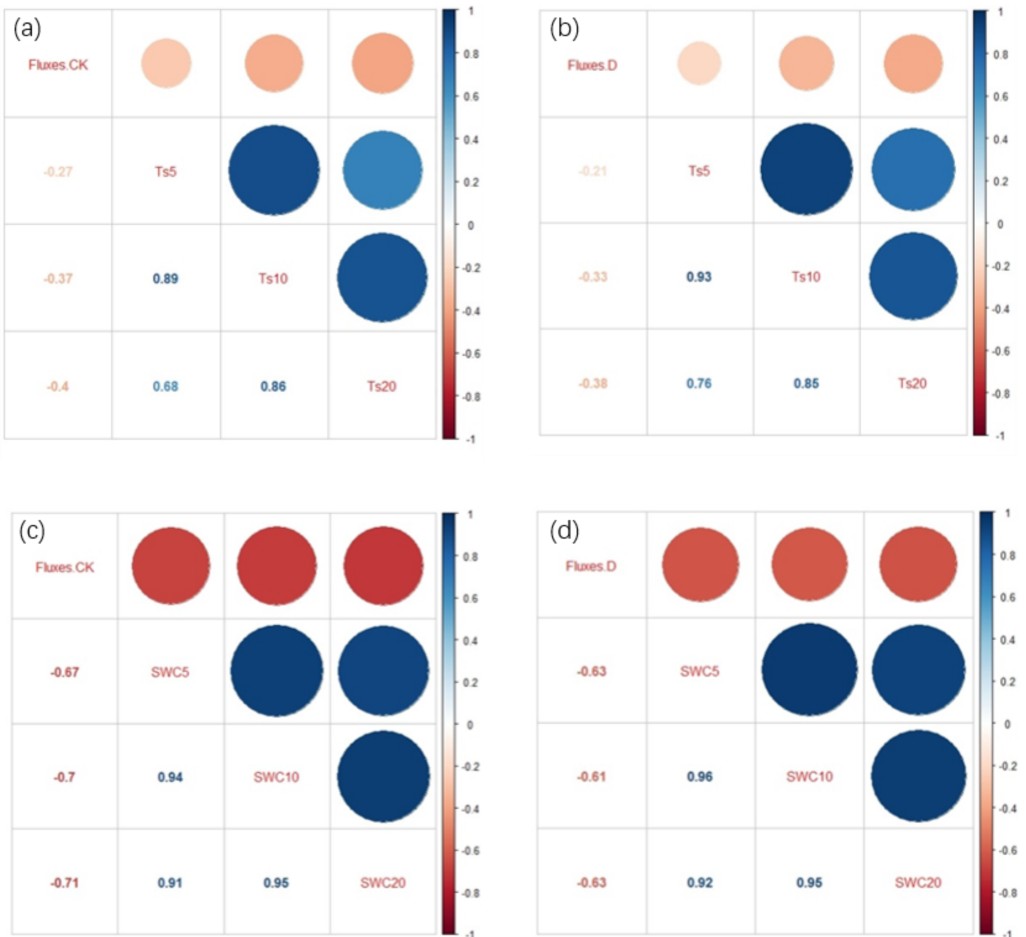

**Figure 6** (A) Correlation coefficient matrix for CH$_4$ fluxes and Ts in the control treatment; (B) correlation coefficient matrix of CH$_4$ fluxes and Ts in the extreme drought treatment; (C) correlation coefficient matrix between CH$_4$ fluxes and SWC in the control treatment; (D) correlation coefficient matrix between CH$_4$ fluxes and SWC in the extreme drought treatment. Fluxes. CK, fluxes in control; Fluxes. D, fluxes in extreme drought; Ts (5, 10, and 20 cm), soil temperature at a depth of 5, 10, and 20 cm; SWC (5, 10, and 20 cm), soil water content at a depth of 5, 10, and 20 cm. Light and dark red represent the degree of negative correlation. Light and dark blue represent the degree of positive correlation.

and the relationship between CH$_4$ fluxes and environmental factors in a typical alpine peatland. The results clearly showed that extreme drought significantly decreased the CH$_4$ fluxes of the peatland ecosystem (Fig. 3B), which was consistent with previous studies (*Goodrich et al., 2013*; *Brown et al., 2013*; *Malone et al., 2013*; *Korres et al., 2017*). With the decrease of SWC and anaerobic degree, the transition from anaerobic environment to aerobic environment decreased the generation of methane and increased the thickness of the oxide layer, and the produced methane was oxidized by more methanogens (*Smith et al., 2003*; *Webster et al., 2012*; *Tiemeyer et al., 2016*). Extreme drought can also decrease the anaerobic environment of CH$_4$ production and reduce the activity of methanogenic

bacteria and anaerobic microsites, thus, decreasing the emission of $CH_4$ (*Tiemeyer et al., 2016*; *Tian et al., 2012*; *O'Connell, Ruan & Silver, 2018*).

Extreme drought also had potential effects on different soil physical and chemical properties (*Fenner & Freeman, 2011*; *Yuste et al., 2011*; *Smith et al., 2015*; *Jiang, Wang & Dong, 2010*). Across the observed content of the soil organic matter, our results indicated that the soil content of TN, TC, and SOC in the control treatment were higher than that under extreme drought (Figs. 4D–4F). As previously reported, one possible explanation for this observation is that drought might alter the distribution and transformation of carbon in the soil via the movement of water and solutes through the pore matrix; thus, this might result the decrease of these matters (*Smith et al., 2015*). Additionally, with the vegetation coverage up to 90% and abundant rainfall during the growing season in the Zoige alpine peatland, the large amount of methanol released from dead plants will provide the substrate for methanogens, but the active conditions for methanogens changes with the changing water conditions of the alpine peatland, resulting in reduced $CH_4$ emission (*Jiang, Wang & Dong, 2010*). Our results also found that the soil contents of TN, TC, and SOC of the subsoil (20 cm) were lower than that of the topsoil (5, 10 cm) (Figs. 4D–4F). In contrast, our results also showed that there was a higher belowground biomass in the subsoil (Fig. 4B) than the topsoil. Moreover, a higher SWC in the subsoil (20 cm) was found relative to the topsoil (Fig. 4C), and this might have been because plants will allocate more roots to absorb more water and nutrients in deeper soils, thus leading to a decreased SWC and soil organic matter (*Johnson et al., 2014*).

Some prior studies have reported that environmental factors, including Ts and SWC, might influence $CH_4$ fluxes (*Morishita, Hatano & Desyatkin, 2003*; *Wei et al., 2012*; *Krause, Niklaus & Schleppi, 2013*). Across the study period, our results found that Ts had a significant negative relationship with $CH_4$ fluxes under control treatments at depth of 10 and 20 cm in the Zoige peatland ecosystem, with the $CH_4$ fluxes decreasing with the increasing of Ts (Figs. 5B–5C). This negative relationship was in agreement with several studies (*Conrad, 1996*; *Butterbach-Bahl & Papen, 2002*; *Koch, Tscherko & Kandeler, 2007*), which suggested that $CH_4$ oxidation rates increased faster with increasing temperature when compared to $CH_4$ production, leading to the decrease of $CH_4$ fluxes. In addition, the alpine peatland is low-temperature and anoxic all year round, but the oxygen content and temperature are increased greatly in the peak period of plant growth, which provides an environment for methane oxidation and enhances the activity of methane oxidative bacteria (*Wei et al., 2012*). Additional results from this study indicated that the correlation between subsoil (20 cm) Ts and $CH_4$ fluxes was better than with the topsoil (5, 10 cm) Ts under these two treatments. This might have been due to the subsoil not being easily disturbed by changes in the external environment, making it more suitable for the survival of microorganisms related to methane production and oxidation (*Koch, Tscherko & Kandeler, 2007*). Another founding in this research was that extreme drought decreased the correlation of $CH_4$ fluxes and Ts (Figs. 6A–6B). One possible explanation for this could be that extreme drought releases sulfate into the soil solution, and this increase could stimulate sulfate-reducing bacteria, which could compete with methanogens for access to

organic substrates that might sever to reduce the influence of Ts on $CH_4$ fluxes (*Dowrick et al., 2006*).

In addition to Ts, $CH_4$ fluxes are sensitive to the SWC, and previous studies have shown a strong relationship between the water table and $CH_4$ emissions (*Dowrick et al., 2006*). Here, we compared the relationship between $CH_4$ fluxes and the SWC at different depths and found that there was a significant negative relationship between $CH_4$ fluxes and SWC in the Zoige peatland ecosystem (Fig. 5). This might have been due to the increase of SWC hindering the diffusion of $CH_4$ into soil pores (*Luan et al., 2018*). By comparing the slope of $CH_4$ fluxes under extreme drought and control treatments, we found that extreme drought significantly decreased the sensitivity of $CH_4$ fluxes towards the SWC (Figs. 5D–5F). A possible explanation for this could be that extreme drought significantly decreased the SWC and changed the hydrothermal conditions of the soil, which could affect the production and oxidation of $CH_4$ fluxes (*Borken & Matzner, 2010*; *Wu et al., 2010*). $CH_4$-oxidizing microorganisms are able to be retrained under extreme drought conditions, resulting in a higher $CH_4$ consumption during a drought, which could lead to the observed decreased sensitivity (*Einola, Kettunen & Rintala, 2007*). In addition, we found a better correlation between $CH_4$ fluxes and subsoil SWC than for topsoil (Figs. 6C–6D). This might be due to the correlation of $CH_4$ emissions and the concentration of $CH_4$ dissolved in the pore water, which was controlled by rhizospheric oxidation of $CH_4$ driven by plant photosynthesis (*Ding, Cai & Tsuruta, 2004*). With more water, the subsoil could provide a beneficial environment for higher methanogen activity (*Tian et al., 2011*). However, a detailed analysis of the microbes and enzyme data is needed to explore these possible mechanisms in the future studies.

## CONCLUSIONS

We found that the condition of extreme drought significantly decreased the $CH_4$ fluxes in the Zoige peatland on the Tibetan Plateau. The Ts and SWC had negative relationships with $CH_4$ fluxes under the extreme drought and control treatments. Extreme drought decreased the correlation of the $CH_4$ fluxes relative to the SWC and weakened the sensitivity of $CH_4$ fluxes towards the SWC. The correlation coefficient between the subsoil (20 cm) environmental factors and $CH_4$ fluxes were higher than it was with the topsoil (5, 10 cm) environmental factors under the extreme drought and control treatments. These findings indicated that extreme drought might reduce the contributions of $CH_4$ emissions from high-altitude peatland into the atmosphere and decrease the global warming potential. However, the mechanism of $CH_4$ fluxes affected by extreme drought remains unclear. As such, our further work will focus on the response of soil enzyme activity and soil microorganisms to extreme drought events and the coupling of microbial process and macroscopic phenomenon.

### Funding

This work was supported by the National Nonprofit Institute Research Grant (CAFYBB2017QB009), the National Key Research and Development Program of China (Grant No. 2016YFC0501804), and the National Natural Science Foundation of China (Grant No. 41701113, 41877421, 31770511). The funders had no role in study design, data collection and analysis, decision to publish, or preparation of the manuscript.

### Grant Disclosures

The following grant information was disclosed by the authors:
National Nonprofit Institute Research Grant: CAFYBB2017QB009.
The National Key Research and Development Program of China: 2016YFC0501804.
The National Natural Science Foundation of China: 41701113, 41877421, 31770511.

### Competing Interests

The authors declare there are no competing interests.

### Author Contributions

- Haidong Wu, Liang Yan, Yong Li and Kerou Zhang performed the experiments, analyzed the data, prepared figures and/or tables, authored or reviewed drafts of the paper, and approved the final draft.
- Yanbin Hao, Jinzhi Wang, Xiaodong Zhang, Zhongqing Yan and Yuan Zhang performed the experiments, authored or reviewed drafts of the paper, and approved the final draft.
- Xiaoming Kang conceived and designed the experiments, authored or reviewed drafts of the paper, and approved the final draft.

### Field Study Permissions

The following information was supplied relating to field study approvals (i.e., approving body and any reference numbers):

Field experiments were approved by the Institute of Wetland Research (project number:20140201).

### Data Availability

The raw measurements are available in the Supplemental Files.

### Supplemental Information

Supplemental information for this article can be found online at http://dx.doi.org/10.7717/peerj.8874#supplemental-information.

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
