# Peer review of "Drought-induced reduction in methane fluxes and its hydrothermal sensitivity in alpine peatland"

_PeerJ, doi:10.7717/peerj.8874_

## Round 0.1 · original submission · Minor Revisions

Dear Dr. Wu and co-authors,

I just received reviews of your manuscript. Although both reviewers consider the study very interesting and providing new findings on the topic, some issues need to be considered before the acceptance.
While reviewer#1 highlights the need to put the importance of your results in the context of global peatlands and methane emissions, reviewer#2 identified numerous errors or inconsistencies that must be corrected. Please, consider all comments and suggestions provide by both reviewers during the revision of your manuscript. Comments from the reviewer#2 have been included in a PDF file.

A comprehensive revision of the English of the manuscript is necessary before submitting the new version.

Don't forget to include a letter response along with the revised version of the manuscript. In this letter you must respond point by point to each question.

Best regards,

Salva

Reviewer 1 ·

Basic reporting

This manuscript is well organized and present CH4 fluxes values from alpine peatlands in extreme drought conditions, however, considering that the main contribution to knowledge is how alpine peatlands respond under that conditions, it is not clear the importance of alpine peatlands or extreme drought events in the global context. The authors would put in context how representative alpine peatlands in worldwide are, and what is the incidence of extreme drought in peatlands. This would serve to make more interesting this manuscript.
I urge the authors to have a native English speaker edit the manuscript.
I would recommend the authors link 'Results' and 'Discussion' into one unique section to make manuscript more fluent.
I recommend improve the references, i.e. in line 50, I think that references 9-10 could be betters.
The authors no provide raw data from flux calculation.

Experimental design

Why the authors use transparent and dark chamber?
How do the authors ensure that the flux calculation is correct with only 2 minutes of measurement?
The authors take in to account the increasing of volume of the chamber due to length of pipe to calculate CH4 flux? The omission of that volume could underestimate the flux values up to a 20%, considering an inner pipe diameter of 2cm.

Validity of the findings

This manuscript offers nothing more than flux values. I feel the authors should find a way to present their findings in a more interesting manner so that they add more than simply fluxes and their relationship with some environmental parameters. As I mentioned previously, the authors could put in context how representative alpine peatlands are to other peatlands.

Additional comments

Some lines where the manuscript could be improved are 169, 172, 184, 185, 285.
I would recommend the authors link 'Results' and 'Discussion' into one unique section to make manuscript more fluent.
The Figure 5a does no show the R2 value.
The use of words “box” and “chamber” (lines115-116) is very confusing. The authors could be clearer in that section.
The influence of SWC appears twice, in line 172 and 181.
The correlation of SWC with CH4 flux could be synthetized, due to in both conditions and all depth analyzed had the same correlation.

Reviewer 2 ·

Basic reporting

no comment

Experimental design

no comment

Validity of the findings

no comment

Additional comments

no comment

Annotated reviews are not available for download in order to protect the identity of reviewers who chose to remain anonymous.

---

## Round 0.2 · accepted · Accept

Dear Dr. Wu and co-authors,

I just want to inform you that the changes made to your manuscript have substantially improved your work and it is now acceptable to be published in PeerJ.

Congratulations!

Salva

Reviewer 1 ·

Basic reporting

no comment

Experimental design

no comment

Validity of the findings

no comment